# Three-dimensional localization and tracking of chromosomal loci throughout the *Escherichia coli* cell cycle
Praneeth Karempudi [1,2], Konrad Gras[1,2], Elias Amselem [1], Spartak Zikrin[1], Dvir Schirman [1] & Johan Elf [1] ✉

The intracellular position of genes may impact their expression, but it has not been possible to accurately measure the 3D position of chromosomal loci. In 2D, loci can be tracked using arrays of DNA-binding sites for transcription factors (TFs) fused with fluorescent proteins. However, the same 2D data can result from different 3D trajectories. Here, we have developed a deep learning method for super-resolved astigmatism-based 3D localization of chromosomal loci in live *E. coli* cells which enables a precision better than 61 nm at a signal-to-background ratio of ~4 on a heterogeneous cell background. Determining the spatial localization of chromosomal loci, we find that some loci are at the periphery of the nucleoid for large parts of the cell cycle. Analyses of individual trajectories reveal that these loci are subdiffusive both longitudinally (x) and radially (r), but that individual loci explore the full radial width on a minute time scale.

The characterization of the movement of chromosomal loci over the *E. coli* cell cycle using fluorescent repressor operator systems (FROS) was pioneered by Lau et al.[1] who inserted 240 repeats of *tetO* and *lacO* close to the *oriC* and *ter* loci and followed them over the replication and division cycle using fluorescent TetR and LacI proteins. More recent work by Cass et al. shows how selected chromosomal loci are organized over the replication cycles[2]. They used a ParB/*parS*-marker in ten different positions on the chromosome to determine how the 1D order of the selected loci corresponds to their cell-long axis localization over the replication cycle. The data support a model where replication occurs in the middle of the cell. After initiation the newly replicated origin of replication, *oriC*, is translocated to the quarter positions along the long axis. Consecutively replicated loci are positioned between the *oriC* in the quarter position and the terminus region at mid-cell. At the time scale of the cell cycle, the chromosomal loci display a seemingly deterministic movement along the cell's long axis to a new position in the cell[3].

The deterministic movement of the average loci position on the generation time scale comes with a more random movement on the 1–100 s time scale. For example, Weber et al.[4] characterized the subdiffusive behavior of 6 chromosomal loci using the ParB/*parS* system. They found that the motion is best described by fractional Langevin motion arising from a combination of the polymer physics and the viscoelastic nature of the cytoplasm. They also observed that the dynamics can be changed by limiting ATP in the cell[5], which suggests that the movement is not only thermally driven.

Although several studies have characterized how individual loci move and localize along the bacterial long axis, far less is known about their 3D location in the bacterial cells. Single-molecule localization microscopy in 3D can be achieved by encoding the z-position of a fluorescence emitter into the point spread function (PSF)[6]. Popular PSF-engineering methods include the astigmatic PSF, double-helix PSF[7], saddle-point PSF[8], and the tetrapod PSF[9]. The first challenge for chromosomal loci imaging with PSF engineering is that the fluorescent label needs to be small compared to the diffraction limit of the microscope used. Unfortunately, the ParB/*parS* and 240x*lacO* markers, which spread over several kb, do not necessarily satisfy this requirement[10]. In addition, these labeling systems have been reported to have physiological effects[1,11,12], which is why we have instead worked with a smaller operator array based on twelve *malO* sites[13] and used MalI-YFP for labeling. The challenge of the relatively short array is a weaker fluorescent signal from the bound proteins, resulting in a signal that is close to the cellular background. Additionally, unbound fluorescent MalI-YFP will contribute to this background signal. The background is heterogeneous and varies on the same spatial length scale as the signal from the molecules of interest, complicating their localization further. This is an important difference compared to previous 3D localization studies of fluorescent proteins in bacteria, which were performed using yellow photoactivatable fluorescent proteins with near-zero fluorescent cellular background, e.g., ref. 14. Photoactivatable proteins can however not be used to continuously track the chromosomal array since only a small fraction of the individual activated

[1]Department of Cell and Molecular Biology, Science for Life Laboratory, Uppsala University, Husargatan 3, Uppsala, Uppsala, Sweden. [2]These authors contributed equally: Praneeth Karempudi, Konrad Gras. ✉e-mail: johan.elf@icm.uu.se

fluorophores would be bound to the array. This problem could in principle be overcome by very strong binding sites and stoichiometric expression of the labeled protein, but under such conditions, constitutively fluorescent proteins would have no background either.

In this work, we address the challenge of localizing chromosomal loci in *E. coli* in 3D using deep learning and data-driven simulations. We used a mother-machine-based fluidic device with open-ended growth channels to trap and image growing *E. coli* for extended periods[15]. To localize fluorescent emitters, a neural network based on field-dependent DeepLoc (FD-DeepLoc)[16] was trained using cubic spline approximation of the experimental astigmatic PSF and heterogeneous cell backgrounds obtained from experimental data. We also calibrated our sCMOS camera to obtain pixel-dependent camera noise models. In summary, we developed a high-precision 3D localization algorithm that can be used at low signal-to-background levels and applied this algorithm to localize chromosomal loci throughout the *E. coli* cell cycle.

## Results

### Simulating ground truth data and training the field-dependant neural network

Deep neural network-based methods are state-of-the-art for 3D localization of fluorescent emitters with high precision for low signal-to-background ratios[16–18], but they require training data that accurately represents the data on which they will be applied. Thus, to localize chromosomal loci in live *E. coli* cells, we trained a field-dependent deep learning model (FD-DeepLoc) from ref. 16, using simulated images of emitters on simulated cell backgrounds closely matching the experimentally observed fluorescently labeled loci (SI sections A.1–A.3).

To simulate the emitter, we used a cubic spline to parameterize the astigmatic PSF[19,20] fitted to images of fluorescent beads to enable sampling at different xyz-positions (Supplementary Fig. 4a). The images were acquired with a custom optical setup[3] equipped with a cylindrical lens in the optical path (Materials and Methods). The choice of an astigmatic PSF, and not a PFS with a longer z-range, was made because of the relatively high background. This PSF confines the light to a smaller imaging area compared to other PSFs and is therefore less hidden in the background. As we were interested in the 3D positions of chromosomal loci, the limited z-range provided by astigmatism was not a major concern since it still covers the width of the nucleoid. Additionally, we rotated the cylindrical lens by 45 degrees to avoid that the astigmatic PSF expansion coincides with the cell's long axis, which otherwise impacts the network's ability to separate the cell background signal from the astigmatic PSF. The simulated emitters were placed randomly in cell-like geometries and simulated astigmatic emitters were sampled from the spline model (Supplementary Fig. 1). The range of simulated gray level values corresponding to the emitters was chosen to match experimental images (SI section A.1).

The background model is based on the background fluorescence in living *E. coli* cells with a labeled chromosomal locus. Phase-contrast images were acquired every minute, and the fluorescence images were acquired every 2 min while the bacteria were growing in a microfluidic mother machine-type device[15]. The locus labeling was achieved by introducing an array of twelve MalI-binding operator sequences at the selected chromosomal locus and expressing a yellow fluorescent protein (YFP) fused with the DNA-binding protein MalI[3]. The MalI-YFP not bound to the operators contributed to the fluorescence background. To obtain a model for the background, the cells were segmented in phase-contrast (Supplementary Fig. 1a, b) and the background fluorescence signal was modeled as a gamma distribution with an average and standard deviation that depended on the distance to the cell boundary (Supplementary Fig. 3). To avoid including pixels corresponding to the operator array, the 25% brightest pixels were omitted in each cell before fitting the gamma distributions (Supplementary Fig. 2). To obtain the simulated ground truth training data, the sampled emitters were overlaid on simulated cell backgrounds (SI sections A.1–3). These images were also passed through the sCMOS camera noise model (SI sections A.4–5) to generate final images in gray-level values.

The simulated data were used to train a FD-DeepLoc model[16] on small image tiles (128 × 128 pixels) with a few simulated emitters present in each cell. The 128 × 128 image size was chosen compared to 40 × 40 pixels in DECODE[17] to include more than one cell, so that the model learns variations in cell background in the FOV. We used FD-DeepLoc as it offers better training stability and convergence properties compared to DECODE. The FD-DeepLoc model also provides features to model field-dependent aberrations and estimate pixel-wise noise over the image. However, since the image tiles used for training of our model were relatively small, we only used the noise estimation features of FD-DeepLoc.

The deep neural network outputs 10 images (channels) of the same size as the input image which quantifies the network predictions. The output channels include one for the probability of an emitter in the corresponding pixel, two for sub-pixel offsets from the center of each pixel, one for axial distance, one for predicted photon counts, four channels for uncertainty in the localizations and photon counts, and one channel for background-free PSFs. Deviation from the known ground truth, quantified by the loss function, was back-propagated to learn the weights in the network over many training epochs until convergence. The model architecture, training, loss functions, and performance of the network are described in Materials and Methods, SI section A.6 and in Supplementary Figs. 6, 7.

### Mapping 3D loci positions to internal cell coordinates

Our trained model was applied to microscopy images of fluorescently-labeled chromosomal loci, with the images being tiled and passed to the network for localization of the emitters (Fig. 1a). Three loci on the chromosome were labeled, one near the origin of replication (*oriC*), one near the terminus (*ter*), and one midway between the origin and terminus (midway), approximately 1.1 Mbp from *oriC* on the right chromosome arm. The corresponding phase-contrast images were segmented using the Omnipose network[21] to generate segmentation masks (Fig. 1b). As the fluorescence and phase-contrast images were acquired with two different cameras, a transformation of emitter coordinates onto the phase-contrast image coordinates was required. Transformed emitter positions in each image were overlaid on the corresponding cell segmentation mask, and the XY coordinates of the emitters were converted to internal long- and short-axis coordinates based on a fit of the cell backbone[13] (Fig. 1c). However, the internal z coordinate could not be assigned in the same way due to a lack of a common reference on the z-axis. The z predictions from the localization network were relative to the focal point of the PSF model (SI section A.3) which was defined as zero. The zero is absolute in relation to the focal locking mechanism of the microscope but this mechanism only looks at a single point of the imaging plane. Thus, if the image plane is tilted, a depth comparison of different parts of the image will be wrong. To correct for a potential tilt, we used the acquired images for each position on the microfluidic chip to get the average z distribution over the imaging plane. We then fit a 2D plane to the average z coordinates of the localized emitters and offset the z coordinates relative to this plane to compensate for the tilt of the sample on the nm scale. Emitter locations from cells imaged over several cell division cycles were pooled to obtain location distributions in x, y, and z (Fig. 1d).

### Determining The Z-range in which the 3D localization precision is stable

Variations of localization precision across the z-range have a direct impact on our interpretation of locus localization inside cells. It is desired that the localization algorithm has uniform precision across the z-range so that each locus localized is represented with equal uncertainty. To this end, we estimated the localization precision of our localization method using fluorescent 100 nm size beads. The beads were imaged repeatedly for 50 frames (Fig. 2a) at different z heights and localized using a network trained with a uniform background.

The localization precision depends on the signal-to-background ratio of the emitter. Thus, for the bead measurements to be representative of locus localization precision, the exposure time was adjusted such that the signal-to-background ratio of the beads (~4.53) would approximately match the

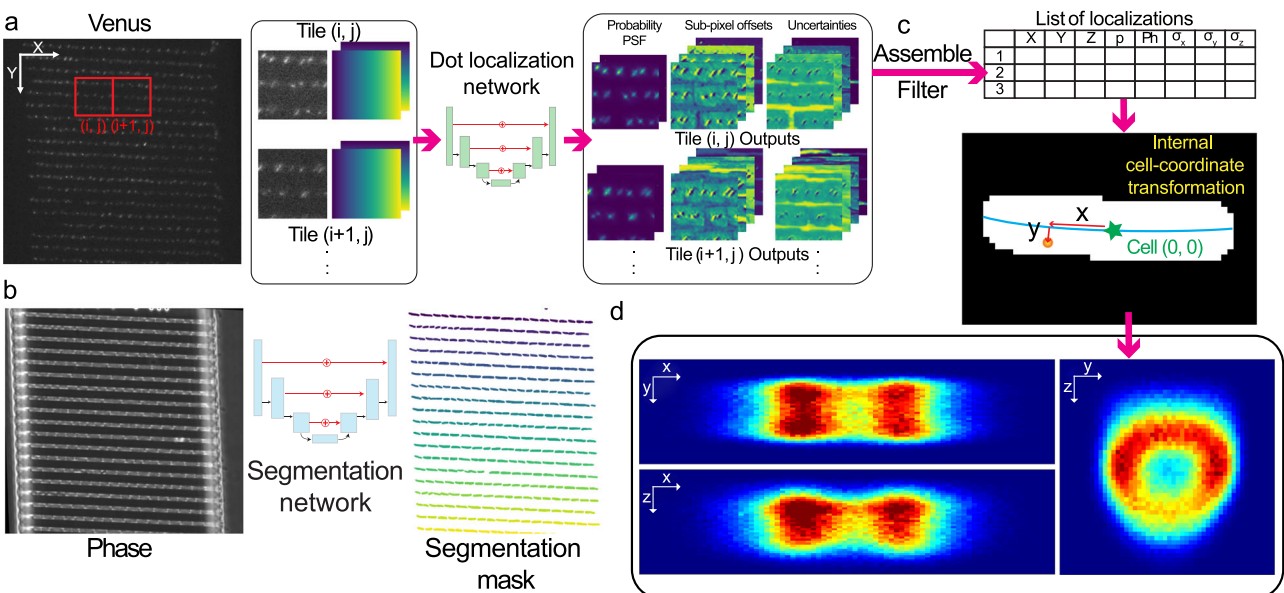

**Fig. 1 | Localization of fluorescent emitters using a deep neural network. a** The fluorescence image is partitioned into tiles together with the pixel location field, i.e., gradient images with the information of pixel location, and fed to a neural network. The network outputs 10 images for each tile (one for the probability of finding an emitter in a pixel (p), one for the cell-background-free PSF (PSF), four images for sub-pixel-x, y, z (x, y, z), and photon counts (ph), and four for their corresponding precision predictions ($\sigma_x$, $\sigma_y$, $\sigma_z$, $\sigma_{ph}$). **b** Phase-contrast image segmented using the Omnipose neural network to produce segmentation labels for each cell. **c** Top:

Localization network outputs from each tile of the fluorescence channel image are assembled and filtered to include emitters $p > 0.8$ to generate the final coordinates of the emitters with respect to the origin of the image. Bottom: Emitters are assigned to cells using a cell segmentation mask and their coordinates are converted to the cells' internal coordinates with respect to their origin (green star) and backbone (blue curve). **d** Localizations from time-lapse imaging pooled together from different cells and plotted as the location distribution of the emitters.

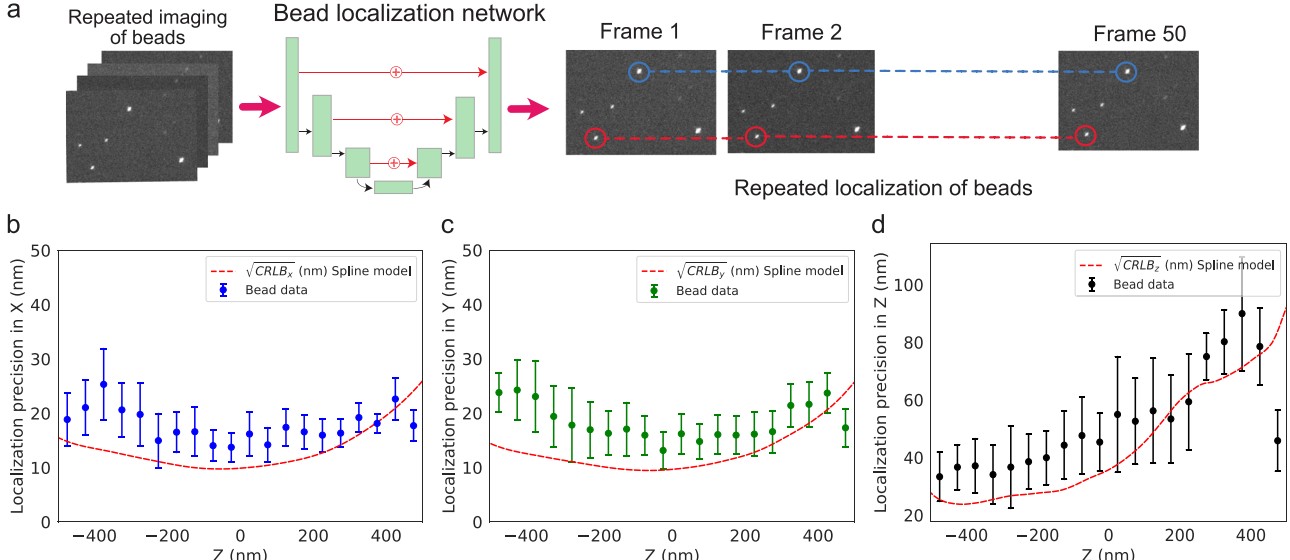

**Fig. 2 | Localization precision as a function of z. a** Scheme for repeated localization of beads. Beads were imaged 50 times per z height and were localized in each image using a neural network. For each bead the standard deviation in their localizations across all frames was used to calculate localization precision as a function of the z height. The signal-to-background ratio of the beads was matched to the signal-to-background ratio of the locus labels (Supplementary Fig. 8). **b–d** Localization precision in x, y, z as a function of z. N = 1311 beads were localized in >30 frames each.

Bead localizations were binned into bins of 50 nm bin size on the z axis and mean and standard deviation of the bin are plotted for x, y and z axis. The red dashed curve in each plot corresponds to the Cramér-Rao lower bound on the localization precision of the cubic spline model of the PSF at a signal of ~1950 photons per emitter and background signal of ~5.8 photons/pixel. The photon count of the detected beads does however vary over the z-range and only the brightest beads were detected reliably (>30 frames) above +300 nm (Supplementary Fig. 9).

signal-to-background ratio observed when imaging labeled chromosomal loci (~3.76) (Supplementary Fig. 8, SI section A.7). The standard deviation of the repeated localizations of stationary beads gives the localization precision. In Fig. 2b–d, we plot the average localization precision in x, y, and z as a function of z as well as the Cramér-Rao lower bound (CRLB) on the

localization precision as calculated from the cubic spline model of the PSF with the corresponding signal. We note that uniform precision can be achieved in the −400 nm to +50 nm range of the PSF (Fig. 2d). These results indicate that, for a trained neural network for emitter localization, imaging in the lower z-range would result in an average emitter localization precision

below 40 nm in all dimensions. Thus, we performed all live-cell imaging experiments with an imaging plane offset of −200 nm in z.

## Consistent 3D localization of fluorescent emitters across the cell cycle

We investigated the positions of three chromosomal loci as a function of cell size to follow them over the cell cycle. 3D location distributions based on the output from the neural network were visualized as heat maps over the cells' long (x-), short (y-), and depth (z-)axes (Fig. 3). The long axis and short axis-location distributions for all loci were similar to previously reported location patterns[2,3,22,23], while the positions on the depth axis were less straightforward to corroborate since they have not been observed before. However, based on the radial symmetry of *E. coli* cells, we expected the location distributions in y and z to be identical. In line with this assumption, the measurements did confirm that the distributions of y and z coordinates show a large degree of similarity. When the distributions were plotted in the yz-plane (Fig. 3), all loci exhibited distinct radially symmetric location patterns.

## The 3D localization error contributions from pooling data from different cells

The *ter* label has a relatively confined localization in the cell center in all dimensions, while the *oriC* label exhibits a broader radial distribution. The *oriC* distribution is however not as broad as the distribution of the midway label, which has a bimodal short-axis localization distribution throughout the division cycle that corresponds to a clear ring distribution in the yz-plane (Fig. 3b). This radial pattern implies that the locus is located close to the edge of the nucleoid, possibly due to interactions with the cell membrane[24] or proximity to highly expressed genes[25]. The ring-shaped distribution of the midway locus gives us an opportunity to estimate the upper bound of the combined errors from localizing the FPs and pooling data from different cells. To estimate this combined error, we assume that the real localizations lie on a perfect circle and are the same in all cells, then the broadening of the circle that results in the ring distribution in Fig. 3b implies that the standard deviation of experimental error is at most 130 nm, as this is the standard deviation of the distribution.

To investigate the localization accuracy in the yz-plane at the single-cell level, i.e., that does not depend on broadening due to pooling data from different cells, we performed a PALM experiment with the membrane protein LacY labeled with photoactivatable PA-mCherry. We found that the localizations of single membrane proteins follow a circle sector in the yz-plane of the cells (Supplementary Fig. 11), as expected for molecules at the cell membrane. However, we were not able to capture the full cell width due to the limited z-range provided by the astigmatic lens and low signal-to-background ratio (~2.56). The standard deviation of the minimal distance to the circle is 69 nm.

## 3D tracking of single loci demonstrates subdiffusive movement radially and longitudinally

The width of the distributions in Fig. 3 could be a result of static differences in locus locations between cells or that the loci explore the whole distribution in individual cells but at different points in time. Using our 3D localizations, we tracked individual loci in time and calculated the mean squared displacement (MSD) as a function of time lag (Fig. 4). In addition to tracking loci on the cell cycle time scale, we also performed tracking experiments on the 10 s time scale (Materials and Methods). MSD estimates from both second and minute time-lapse experiments were plotted together in a log-log plot as a function of time lag to investigate the 3D diffusion of the locus across different time scales (Fig. 4a, b). In line with previous observations[4], we observed sub-diffusion with an alpha = 0.4 along the long axis of the cell, corresponding to the log-log plot slope, assuming the simple model $MSD(\tau) = 2D\tau^\alpha$. However, if we also take the localization error into account as $MSD(\tau) = 2D\tau^\alpha + \varepsilon$, alpha = 0.53 (Supplementary Fig. 13)

along the long axis. Using the tracked y and z coordinates of the midway locus, we also estimated MSDs on the radial axis (Fig. 4b). In the radial direction, we observed sub-diffusive movement up to a plateau corresponding to the width of the ring-shaped distribution in the yz-plane (Fig. 4d). This indicates that individual loci are confined to move in the ring (Fig. 3b), avoiding the center of the nucleoid, and that loci in individual cells explore the full radii of the ring on a time scale of minutes (SI section A.10). The long-axis MSDs, on the other hand, did not exhibit any clear plateau, but approached the full width of the population-based distribution on the 20-min time scale (Fig. 4a, c). On a shorter time scale, up to a few minutes, the average locus locations in different cells can be quite different (Fig. 4c).

Using the same y and z coordinates as for the radial MSD, we also estimated MSD along the cell's angular axis (Fig. 5a). The angular MSDs demonstrate that the trajectories are seemingly free to move around the nucleoid. This could, in principle, be due to the rotation of the cells in the microfluidic chamber, in particular since many trajectories appear directional in the angular coordinate (Fig. 5b, Supplementary Fig. 14). For a small fraction of the cells, we can follow the movement of the same locus on two sister chromosomes in the same cell (Fig. 5c). Calculating the correlation between the angular movement of both locus copies in the same cell (Fig. 5d) results in a very broad distribution. However, there is a tendency towards a positive correlation indicating an average net rotation, which would contribute to the increase in angular MSD at long time scales.

Finally, the localization precision for the chromosomal loci in living cells can be estimated from the extrapolated MSD at time lag = 0, which corresponds to the $2\sigma^2$, which gives $\sigma_x = 45$ nm, $\sigma_y = 32$ nm, $\sigma_z = 57$ nm, and $\sigma_r = 52$ nm (Supplementary Fig. 15). In this estimate, the data in one trajectory is only related to itself and does not include errors from merging data from different cells in the same coordinate system or cells at different heights in the microfluidic trap.

## Discussion

3D localization of chromosomal loci with small FROS markers in bacteria has been challenging mainly because of the low signal-to-background ratio originating from molecules not bound to the operator arrays. Even with the strongest binding sites, the long search times before binding, as compared to the generation time of the cell, make background fluorescence detrimental without excessively long operator arrays.

To achieve accurate super-resolved 3D localization of emitters, we build on the FD-DeepLoc algorithm. Our most significant addition to the method is obtaining ground truth training data for its application to emitters with low signal-to-background ratios. This is critically needed for the algorithm to give meaningful results. Importantly, we find that the cell background varies depending on the distance to the cell boundary. Due to the low signal-to-background ratio regime of the experiments and the similar spatial extension of cell background and emitter PSF, accounting for the structured background is essential for the effective use of localization methods and the generation of training data.

To test the functionality of the overall 3D localization method, we determined the distributions of different chromosomal loci over the cell cycle. Since the *E. coli* cell is radially symmetric, we expected the same spatial distribution of emitters along the width and depth of the cell. We could show this radial symmetry for all chromosome labels. Interestingly, we also observed that some chromosomal loci are confined to the periphery of the nucleoid, which has not been directly observed before. The individual loci move subdiffusively in the radial direction and explore the full width of radial confinement on the 2-min time scale. The subdiffusion along the long axis only approaches the width of the population-based distribution on the time scale of the cell cycle (Fig. 4d), which implies that the width of the distribution is effectively due to averaging over different cells.

With respect to localization accuracy, we have three different types of estimates. (i) The lower limit for the localization error is given by repeated

**Fig. 3 | 3D locus locations over the cell cycle.** 3D locus location distributions throughout the cell cycle visualized using two-dimensional histograms of fluorescent emitter positions along the long, short and depth cell axes from three different *E. coli* strains carrying chromosomal locus labels (**a**), near *oriC*, (**b**), a locus 1.1 Mbp from *oriC* on the right chromosome arm, and (**c**), near *ter*. (Left) Two-dimensional histograms as a function of cell area. (Right) Two-dimensional histograms along long, short and depth cell axes over all detected cell areas. Two-dimensional histograms for the midway locus are the same as in Fig. 1d. The color in each histogram heat map indicates the number of emitters per cell and histogram bin, with all the two-dimensional histograms using the same bin size and bin positions. Distributions of cell outlines detected in phase-contrast are shown in white. In figures with cell outlines, the old cell pole is always pointing up. White dashed lines indicate average birth and division sizes. The average birth and division areas have been rounded to the nearest area bin edge. 3D location distributions for repeated experiments are shown in Supplementary Fig. 12. Statistics can be found in Dataset S1.

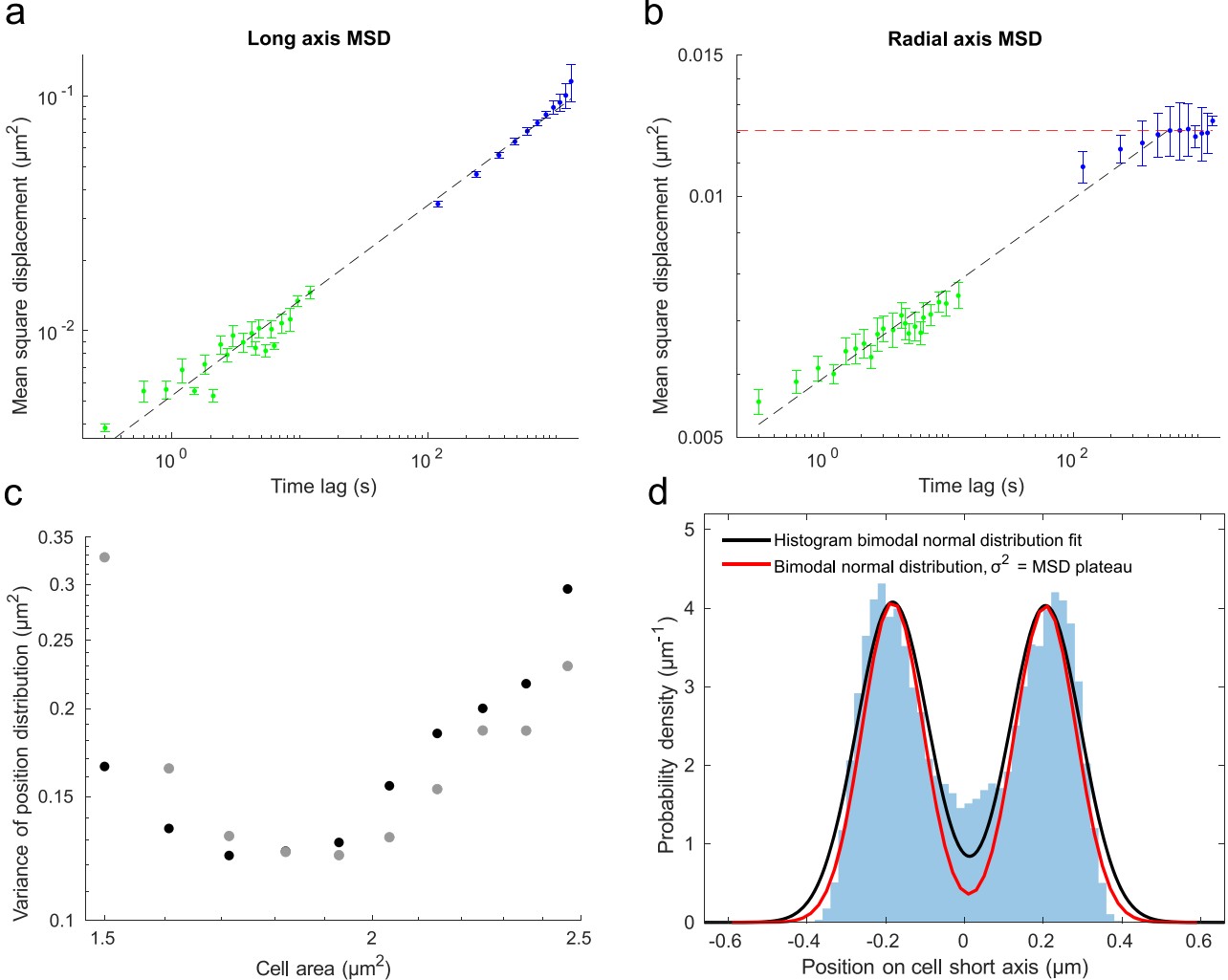

**Fig. 4 | 3D diffusion of a chromosomal locus.** Midway locus MSD as a function of time lag along the cell's (**a**), long and (**b**), radial axes. Data points in green are based on seconds time scale experiments (300–30,000 ms between trajectory frames), and data points in blue are based on minute time scale experiments (2 min between trajectory frames). Data points in green show averages of 5 experiments and data points in blue show averages of 2 experiments. Error bars correspond to standard error of the mean. Black dashed lines show linear fits to the logarithm of the MSD values (Long axis slope (with 95% confidence interval): 0.4049 (0.3872, 0.4226). Radial axis slope (with 95% confidence interval): 0.1116 (0.104, 0.1192)). The red dashed line indicates a plateau estimated as the mean of radial MSD values for the five longest time lags. Trajectory long axis coordinates were set to have the same starting point and their mean movement due to cell growth was subtracted before

MSD estimation. Long axis MSDs estimated without net movement subtraction are shown in Supplementary Fig. 13a. The MSD values are based on >=167 trajectories. **c** Variances of the population-based distribution (Fig. 3b, Supplementary Fig. 12 (Midway)) along the cell's long axis as a function of cell area. Black and gray data points correspond to variance estimates from two different experiments. **d** Histogram from the yz-plane 2D histogram (Fig. 3b) along depth axis bin 0. Black curve corresponds to a bimodal normal distribution fit of the histogram ($\sigma$ = 130 nm). Red curve corresponds to a bimodal normal distribution with the same amplitude and mean as the histogram fit, but with a variance estimated as the mean of the radial MSDs for the five longest time lags ($\sigma$ = 110 nm, red dashed line in Fig. 4b). Statistics can be found in Dataset S1.

localization of the same fluorescent bead at approximately the same signal-to-background ratio as the chromosomal loci, which results in a localization precision of 15–20 nm in x, y and 35–40 nm in z. (ii) An upper limit of the localization error is given by how well the spatial features are resolved in the population-based distributions, such as the width of the ring distribution in the yz-plane for the midway locus. On top of the localization error, cell-to-cell differences, the errors in cell segmentation, and the actual biological width of the structure contribute to this width, which is in the order of 130 nm in r. (iii) The best localization accuracy measurement for individual loci is given by the MSD extrapolation to zero time lag displacements, as calculated from single-locus tracking data. This gives us 45 nm in x, 32 nm in y, 57 nm in z, and 52 nm in r. The radial MSD (Fig. 4b) spanning the range between 52 and 130 nm on the cell cycle time scale shows that the breadth of the radial distribution is due to the width of this structural feature in individual cells and not a broadening due to combining data from different cells.

## Materials and methods
### Bacterial strains
All imaging experiments with bacterial strains were performed in M9 minimal medium with 51 µg/ml Pluronic F-108 (Sigma-Aldrich 542342), 0.4% succinate, and 1× RPMI 1640 amino acid solution (Sigma) at 30 °C. Inoculation of bacterial strains from cryo-stocks into growth medium was performed one day before each experiment. After growing the strains overnight at 30 °C in a shaking incubator (200 rpm), they were diluted 1:100 and loaded in a microfluidic chip. Genotypes of all strains used are described in Table S1.

### Optical setup
Imaging was performed with a Ti-E (Nikon) microscope equipped with a 100X immersion oil objective (Nikon, NA 1.45, CFI Plan Apochromat Lambda D MRD71970) for both phase-contrast and wide field

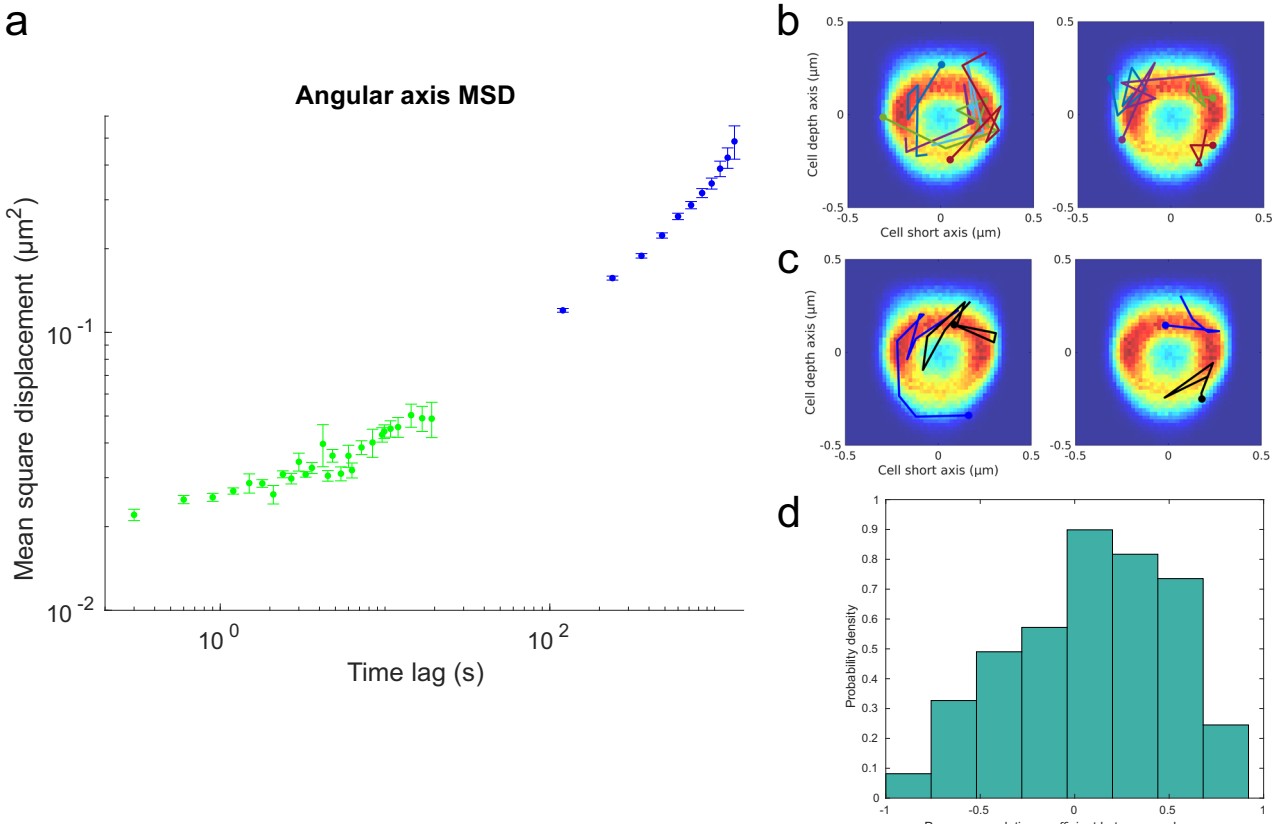

**Fig. 5 | Rotational movement of tracked loci. a** MSD as in Fig. 4a, b but along the cell's angular axis. Trajectory angles (in radians) were multiplied with their corresponding radii to calculate the arc length before the MSD estimation. **b** Selected examples of YZ trajectories and long axis trajectories from time-lapse experiments on the minute time-scale that show rotational or confined diffusion, overlaid on yz-plane location distributions. **c** As in Fig. 5b, but with trajectories in blue and black corresponding to locus copies from the same cell. Trajectories in blue were in the cell half closest to the new pole and trajectories in black were in the cell half closest to the old pole. The yz-plane location distributions are the same as in Fig. 3b. **d** Histogram of Pearson correlation coefficients estimated between yz-plane trajectory angles (in radians) for locus copies from the same cell. The histogram was based on tracking 51 locus pairs.

epi-fluorescence microscopy. Fluorescence images were acquired using a Kinetix sCMOS camera (Teledyne Photometrics), where the camera acquisition triggered a 515 nm laser (Fandango 150, Cobolt) for SYFP2 excitation using a function generator (Tektronix). Images were acquired at 150 ms exposure time and 5 W/cm² power density. The emitted fluorescence was passed through a FF444/521/608-Di01 (Semrock) triple-band dichroic mirror. Fluorescence was passed through Nikon Ti-E microscope tube lens ($f = 200$ mm) to form an imaging plane, a cylindrical lens ($f = 500$ mm, Thorlabs LJ1144L2) was placed 37 mm after that imaging plane before entering a lens relay system including a BrightLine FF580-FDi02-T3 (Semrock) dichroic beamsplitter. The fluorescence was then filtered with a BrightLine FF01-505/119-25 (Semrock) and focused on the camera by the last lens in the relay system. In-house plugins to Micro-Manager were used to run the microscope.

Phase-contrast images were acquired using a DMK 38UX304 camera (The Imaging Source) with a 50 ms exposure time. The light source used for phase contrast was a 480 nm LED and a TLED+ (Sutter Instruments). The transmitted light was passed through the same FF444/521/608-Di01 (Semrock) triple-band dichroic mirror as the fluorescence and reflected before the image plane away from the fluorescence light path using a Di02-R514 (Semrock) dichroic mirror. A lens relay system with a phase ring at the back focal plane was used to direct the light onto the camera.

Experiments performed on the minute time scale (Figs. 3–5) involved acquisition of phase-contrast images every minute and fluorescence images every 2 min at 16 positions on the microfluidic chip sample for each bacterial strain.

Experiments on the seconds time scale (Fig. 5) involved acquisition of 1 phase-contrast image followed by 10–12 consecutive acquisitions of fluorescence images for each position on the microfluidic chip sample. All fluorescence images were acquired with a 150 ms exposure time with different time lags between the fluorescence images for the MSD estimation introduced as non-exposed frames using the SMART Streaming function of the Kinetix camera. The function allows for triggering of acquisitions with different exposure times, allowing all exposed images in a stack to have a 150 ms exposure time and all non-exposed images to have a different exposure time. We used 150, 450, 750, 1050, and 2150 ms exposure times for the non-exposed frames, which were acquired between each exposed frame, to achieve frame-to-frame time lags of 300, 600, 900, 1200, and 2400 ms between exposed frames. Image acquisition was performed for 30–40 positions on the sample per time lag.

PALM experiments with the strain expressing LacY-PAmCherry were performed in a microfluidic device with supply of growth medium. Image acquisition involved acquiring 500 fluorescence images with a 780 × 500 pixel ROI followed by 1 phase contrast image at each position on the microfluidic chip sample. The fluorescence image acquisition involved 30 ms exposures with a 405 nm laser (Cobolt) set at a 25% duty cycle for fluorophore activation and 30 ms exposures with a 580 nm laser (VFL, MBP Communications) for fluorophore excitation.

**Microfluidic chip**

All microfluidic experiments were performed using modified mother machine devices[15], where cells were loaded into the growth channels and

continuously supplied with growth media. Each imaged FOV contained 20 growth channels, corresponding to a 69 μm × 69 μm area.

## Image analysis pipeline

Experiments on the seconds and minute time scales were analyzed with an automated image analysis pipeline[26] that was primarily written in MATLAB. This involved segmentation of phase-contrast images with an Omnipose network[21,27] used to determine outlines of bacterial cells in the images. In the experiments on the minute time scale, the cells were tracked using the Baxter algorithm[28] to assemble cell lineages. No cell tracking was performed in the seconds time scale or PALM experiments. For all experiments, fluorescent emitters were localized in 3D using the neural network described in this study. The emitter's internal coordinates along the long and short axes were estimated by first fitting a cell backbone as a second degree polynomial between the cell poles and then estimating the distance to the cell poles for the long axis coordinate, and the signed distance to the cell backbone for the short axis coordinate. The z coordinates were not transformed as the x and y coordinates due to a lack of reference along the z-axis of the cells. The corrections for variations in the height of the imaging plane were done as described in the main text.

## 3D locus location distributions

Post-processing of localized 3D locus coordinates to visualize them in two-dimensional histograms (Fig. 3) involved pooling of long, short and depth axis coordinate data from all cells that had been successfully tracked for at least 20 min, and that also had mother and daughter cells that had been tracked for at least 10 min. For two-dimensional histograms as a function of the cell cycle, the 3D coordinates were binned on cell areas, based on the area enclosed by the segmented cell outline.

## 3D locus tracking

Post-processing of localized 3D coordinates to assemble them into 3D trajectories was done only for experiments with the midway locus label. 3D trajectories were assembled from emitter coordinates that could be tracked for at least 5 consecutive frames. The coordinates were based on fluorescent emitters localized in the cell half closest to the new cell pole, based on their long axis coordinates. Only emitter coordinates were included from cells that in the corresponding phase-contrast had a cell area >= 1.6 μm$^2$, based on the area enclosed by the segmented cell outline. Selection based on cell area and distance to the new cell pole was performed to ensure that single loci were tracked. For tracking of two locus copies (Fig. 5), the distance between each emitter and the old and new poles was used to separate the trajectories. 3D tracking of emitters minute time scale experiments also required the cells to be tracked for at least 20 min, and for these cells to have mother and daughter cells that were both tracked for at least 10 min.

## Segmentation masks generation

Phase-contrast images were segmented using Omnipose[21,27] cell segmentation networks trained specifically for mother-machine devices. The segmentation network is described extensively in ref. 21. These segmentation masks were used to generate Euclidean-distance transform (EDT) to the boundary field whenever required. Values of EDT were rounded to the nearest integer. Cell masks were used for sampling emitter coordinates when simulated microscopy images were provided as input during localization network training. Conversion of global emitter coordinates with respect to the origin of the image into the internal coordinates of each cell was also done based on the cell segmentation masks.

## PSF calibration and CSpline model fitting

Images of beads (TetraSpeck T7279 100 nm size) immobilized on a cover-slip were imaged, at a high signal-to-noise ratio, in 25 nm steps in z and used to generate a PSF using the SMAP package[29]. About 50 beads were localized and aligned in 3D space using SMAP to generate a mean PSF image stack. All images in the mean PSF stack were also normalized by the maximum intensity value on the mean image of the central three image planes around

the focal point of the 3D PSF stack. After normalization, a piecewise cubic spline model (Eq. 1) was fitted to the normalized PSF stack, giving 64 spline coefficients per voxel. These coefficients were used to interpolate intensity values inside the voxels of a PSF image stack and generate an image for a given emitter that was placed in the simulated microscope image. The PSF intensity variations across the z-range are covered by the network training when sampling emitter intensity, thus sampled PSF images in 2D were normalized to sum to 1 before converting to photons during simulations. The 3D PSF spline model follows the piecewise cubic spline equation:

$$f_{ijk}(x, y, z) = \sum_{m=0}^{3} \sum_{n=0}^{3} \sum_{p=0}^{3} a_{ijkmnp} \left(\frac{x - x_i}{\Delta x}\right)^m \left(\frac{y - y_j}{\Delta y}\right)^n \left(\frac{z - z_k}{\Delta z}\right)^p \quad (1)$$

where $f_{ijk}(x, y, z)$ gives the normalized 3D PSF intensity around the voxel $(i, j, k)$ at the coordinate $(x_i, y_j, z_k)$, $\Delta x$, $\Delta y$ are pixel sizes, and $\Delta z$ is the step height of the bead stack during acquisition, $a_{ijkmnp}$ are their cubic spline coefficients and $(x, y, z)$ is bounded within the interval $(x_i, y_j, z_k)$ to $(x_{i+1}, y_{j+1}, z_{k+1})$ for each voxel $(i, j, k)$. The first and second derivatives of the splines of neighboring voxels are constrained to be equal. Using this function we can interpolate the PSF and retrieve a PSF image at desired z-depth, this can later be placed in the simulated microscope image to represent an emitter at any real-valued location within the simulated image.

## Network architecture and training

The neural network models used in the paper consist of two U-nets connected in series[16]. During training, the input to the network consists of fluorescence images simulated using the cubic spline PSF (SI section A.3) and background estimation models (SI sections A.1 and A.2) and camera coordinate fields defining the sub-region of the camera used at that training step. Emitters were sampled randomly in space with average densities specified per cell. Cell backgrounds for simulations were done using pixel intensity distribution fits described in the SI sections A.2. Background pixels from the fluidic chip were also sampled from similar fitted intensity distributions. Camera field input was given using the CoordConv strategy[16,30] to both U-nets to learn camera pixel-dependent noise. A detailed description of the image simulation is described in the SI section A.1.

All networks output 10 channels of the same size as the input image: one channel for probability $\hat{p}_k$ of emitter in pixel $k$, two channels for sub-pixel offsets $(\Delta\hat{x}_k, \Delta\hat{y}_k)$ from the center of each pixel, one channel for axial distance $\Delta\hat{z}_k$, one channel for photon counts $\hat{I}_k$, four channels for location errors $\Delta\hat{\sigma}_{xk}$, $\Delta\hat{\sigma}_{yk}, \Delta\hat{\sigma}_{zk}, \Delta\hat{\sigma}_{Ik}$ and one channel for the cell-background-free PSF. $\Delta\hat{x}_k, \Delta\hat{y}_k$, $\Delta\hat{z}_k$ predictions were scaled to be in $[-1, 1]$ using the Tanh activation function while $\hat{p}_k$ and $\hat{I}_k$ were scaled to be in $[0, 1]$ using sigmoid activation function. The maximum photon count was set for each model during training, and the same scaling was applied at the prediction time. All predictions were scaled back to real values (nanometers, intensity values) using appropriate scale factors that were applied on the inputs to the network during training. During training emitter photon counts were sampled uniformly in the range [750, 3000] and z values were sampled in $[-700, 300]$ nm.

Networks were trained using the AdamW optimizer with a learning-rate scheduler until convergence using PyTorch 2.0. Evaluation metrics used to estimate localization performance were the same as for the SMLM 2016 challenge[31]. The localization networks were trained for 20000–40,000 iterations with an initial learning rate (0.0006) that was reduced by 90% after every 5000 iterations. The batch size was 64. AdamW optimizer with a weight decay of 0.1 used for training. Metrics for a trained model are shown in the SI section A.6.

## Loss functions

The loss functions used are the same as in FD-DeepLoc[16] but the contributions are reweighted, i.e.,

$$\mathscr{L} = w_{count}\mathscr{L}_{count} + w_{CE}\mathscr{L}_{CE} + w_{loc}\mathscr{L}_{loc} + w_{Mol}\mathscr{L}_{Mol}$$

The first term, $\mathscr{L}_{count}$, relates to the number of emitters in the image. Here, the true number of emitters in an image with $K$ pixels $E = \sum_{k=1}^{K} p_k$, given the Bernoulli probability $p_k = 0 \, or \, 1$ of each pixel. When many emitters are present in an image, this Binomial distribution can be approximated as a Gaussian distribution with mean $\mu_{count} = \sum_{k=1}^{K} \hat{p}_k$ and variance $\sigma_{count}^2 = \sum_{k=1}^{K} \hat{p}_k(1 - \hat{p}_k)$ as $p_i$ is independent of $p_j$ for all our experiments.

$$\mathscr{L}_{count} = -\log P(E | \mu_{count}, \sigma_{count}^2) = \log\left(\sqrt{2\pi}\sigma_{count}\right) + \frac{1}{2} \frac{(E - \mu_{count})^2}{\sigma_{count}^2}$$

Thus, this loss function will penalize deviations from the true amount of emitters.

The cross-entropy term $\mathscr{L}_{CE}$ is a binary cross-entropy between the ground truth pixel-wise probability map $p_k$ and the predicted probability map $\hat{p}_k$:

$$\mathscr{L}_{CE} = -\sum_{k=1}^{K} \left[ p_k \log \hat{p}_k + (1 - p_k) \log(1 - \hat{p}_k) \right]$$

This term penalizes the positioning of emitters in the image that deviates from the true emitter position. The two first terms relate to the precise amount of emitters and their positions per pixel, but are not considering the sub-pixel emitter placement, localization uncertainty or the PSF shape. For the neural network to learn sub-pixel resolution, intensity estimates and localization error, we model a per pixel distance penalization map away from the emitter as a 4D Gaussian, $P(u_e^{GT} | \mu_k, \Sigma_k)$, centered on each true emitter $u_e^{GT} = [x_e, y_e, z_e, I_e]$. The 4D Gaussian can be expressed as

$$P(u_e^{GT}, |, \hat{\mu}_k, \hat{\Sigma}_k) = \frac{1}{\sqrt{(2\pi)^4 \det(\hat{\Sigma}_k)}} \exp\left(-\frac{1}{2}(\hat{\mu}_k - u)^T \hat{\Sigma}_k^{-1}(\hat{\mu}_k - u)\right),$$

where $\hat{\mu}_k = [x_k + \Delta\hat{x}_k, y_k + \Delta\hat{y}_k, z_k + \Delta\hat{z}_k, \hat{I}_k]$ is the pixel coordinate position with a per pixel sub-pixel offset together with the intensity of the pixel and $\hat{\Sigma}_k = diag(\hat{\sigma}_{xk}^2, \hat{\sigma}_{yk}^2, \hat{\sigma}_{zk}^2, \hat{\sigma}_{Ik}^2)$ is a per pixel covariance matrix. To suppress pixels with low probability of having an emitter we weight the Gaussian with the normalized emitter pixel probability. Doing this for each true emitter and summing over them corresponds to the loss function contribution $\mathscr{L}_{loc}$:

$$\mathscr{L}_{loc} = -\frac{1}{E} \sum_{e=1}^{E} \log\left(\sum_{k=1}^{K} \frac{\frac{\hat{p}_k}{K}}{\sum_{j=1}^{K} \hat{p}_j} P(u_e^{GT} | \hat{\mu}_k, \hat{\Sigma}_k)\right)$$

Note that none of the loss function contributions described so far address the PSF shape explicitly. This is handled by the final term. The $\mathscr{L}_{Mol}$ term minimizes the error between simulated PSF and predicted PSF images:

$$\mathscr{L}_{Mol} = \sum_{k=1}^{K} (\hat{I}_k - I_k)^2,$$

where $I_k$ is the true image of noise-free simulated emitters (Supplementary Fig. 1g) including the PSF shape.

Each loss component is multiplied with a weight such that all four loss components have the same order of magnitude. The weights of different loss components ($w_{count}$, $w_{CE}$, $w_{loc}$, $w_{Mol}$) are set to 0.2, 0.25, 1.0 and 8000 respectively, to get the contributions to a similar range as we assume that each part of the loss function has equal importance.

## sCMOS camera model

An sCMOS camera (Photometrics Kinetix) was calibrated using the Accent plugin[32,33] in Fiji to estimate pixel-dependent camera parameters (SI A.1)

such as e-/gray level, read noise, and thermal noise. The camera was operated in sensitivity mode (12-bit) and dark images for calibration were acquired with varying exposure times using a Micro-manager 1.4 script. The same sCMOS sub-region was used during camera calibration and when acquiring bead and cell data. Pixel dependent camera parameters, gain, read noise, and thermal noise, were used to sample noise during the simulation of training data for all neural networks. Due to the noisy estimation of gain values per pixel using this method, the median gain value (~2.58 gray level/e-) was used for conversions between e- and gray level wherever required. Camera parameter distributions are shown in SI section A.5.

## Transformations between phase and fluorescence

Images of fluorescent beads (TetraSpeck T7281 500 nm size) were captured on both cameras. Bead positions were selected manually on both images using the *cpselect* function in MATLAB. Geometric transformations of the projective kind were fitted using the *fitgeotrans* function in MATLAB, similar to the procedures described in ref. 3.

## Inference and Post-processing

Network outputs of experimental images were processed to construct a list of emitters in each image. The probability output channel was used to obtain emitter pixel locations after non-max-suppression as described in ref. 16. Offsets from the center of the pixel and localization precision were obtained by masking the other output channels using the emitter pixel locations. Final emitter locations were converted to a list of coordinates in nanometers from the origin of the image (top left corner).

## Statistics and reproducibility

All experiments were performed using the same setup. No randomization was performed. In the analysis of microscopy image data, cell outlines with transient drops or increases in size were deemed as missegmentations and were not used in the analysis. Cells with large center of mass movements of the cell outlines were excluded from the analysis. Cell lineages with short life span, without a mother or two daughter cells were excluded from the analysis. The results shown for different chromosomal loci were reproducible across 16 different FOVs. Detailed statistical information can be found in the figure legend of each figure.

## Data availability

Raw datasets and Supplementary Data 1, 2 are available in a Figshare repository, accessed at https://doi.org/10.17044/scilifelab.25998934.

## Code availability

Neural network models, and code for 3D distribution plots can be found at https://doi.org/10.17044/scilifelab.25998934.

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

## Acknowledgements
We wish to thank Irmeli Barkefors and David Fange for helpful input on the manuscript and Patrick Hennig for work on background models. We acknowledge funding from SSF (ARC19-0016), the European Research Council (BIGGER:885360), the Knut and Alice Wallenberg Foundation (2016.0077; 2017.0291; 2019.0439) and the eSSENCE e-science initiative to J.E. The computations and data management were enabled by resources provided by the Swedish National Infrastructure for Computing at UPPMAX, partially funded by the Swedish Research Council through grant agreement no. 2018-05973.

## Author contributions
J.E. conceived the project. P.K. developed the method and put together the processing pipeline. K.G. performed all the experiments, analyzed the time-lapse experiments and wrote the 3D tracking code. S.Z assembled the analysis pipeline, wrote the code for image transformations and 3D localization corrections. E.A. built the optical setup and provided feedback during the entire project. D.S. constructed most strains and provided feedback during the project. P.K., K.G., J.E. wrote the paper with inputs from all co-authors.

## Funding

## Competing interests
The authors declare no competing interests.
