## [Transparent Peer Review file · Communications Biology]

Three-dimensional localization and tracking of chromosomal loci throughout the *Escherichia coli* cell cycle

Corresponding Author: Professor Johan Elf

Version 0:

Reviewer comments:

Reviewer #1

(Remarks to the Author)

In this paper, Karempudi et al. present a deep learning method for astigmatism-based 3D localization of chromatin loci in live *E. coli* cells. The authors characterized the PSF, fluorescence background, and system noise, and used these for training the neural network. This method provides the coordinates of fluorescently labeled molecules relative to the cell backbone, using a phase-contrast image of the cell as a reference for the coordinate system. The evaluation of the localization precision on fluorescent beads demonstrated good performance. Using this method, the authors tracked three chromosomal loci at different time scales and across the cell cycle.

The paper presents well-constructed experiments and thorough quantitative analysis. The study is overall well-executed, with the findings supporting the main conclusions. This research will likely capture the interest of a wide audience within the journal's readership. While the work is solid, I do have a few minor concerns that should be considered.

1. In Fig. 2, the rightmost points ($Z \approx 450$ nm) are significantly below the CRSB for X, Y and Z. How can this be explained?
2. Lines 223-235. The description of the localization of the *ter*, *oriC*, and midway loci does not seem to accurately reflect the data in Fig. 3. The authors note that the localization of *oriC* is different from both *ter* and midway; however, in Fig. 3, the distributions of *oriC* and midway appear qualitatively similar to each other (ring-like in ZY and doublet in XY in larger cells).
3. Would it be possible to calculate the MSD for the origin and terminus and compare it with the MSD for midway?
4. Is there any new insight into bacterial chromatin that can be drawn from the data presented in the paper?

Reviewer #2

(Remarks to the Author)

The authors utilized FD-DECODE for the 3D localization and tracking of chromosomal loci in live *E. coli* cells under low SNR conditions. The manuscript is well-prepared, but many details need clarification or correction.

(1) FD-DECODE was developed to address field-dependent aberrations. In Fig. S1, it is unclear if such effects are present in such small regions (128 x 128 pixels). The rationale for training the network on these small regions instead of the full image should be explained.

(2) From the Methods section, it is unclear how the astigmatic PSF was generated. Was a cylindrical lens used for this purpose?

(3) Fig. S4, the astigmatic PSF is oriented at 45 degrees rather than the more commonly used 90 degrees. The reason for this orientation should be provided.

(4) Fig. S4 indicates a significant spherical aberration in the system, which results in asymmetrical z localization precision. This aberration can be corrected by adjusting the correction collar of the objective.

(5) Fig. S5, there are well-established protocols for camera calibration. I don't know why the authors used the Accent plugin. The gain calibration result (Fig. S5A) doesn't make sense to me, which shows negative gain values. I suggest the authors use the gain value provided by the manufacturer or perform calibration following the right protocols (DOI: 10.1038/nmeth.2488; DOI:10.1038/s41467-022-30907-2).

Reviewer #3

(Remarks to the Author)

Localizing chromosomal loci in the Escherichia coli cell cycle in 3D is challenging due to the low signal-to-background ratio, and the structured background is essential for the localization and the generation of training data. To overcome this challenging, this manuscript proposed and developed a field-dependent neural network based on Fu et al. (2022)'s field-dependent DeepLoc algorithm. The contributions of this manuscript include simulating ground truth training data and 3D localization with high localization accuracy.

However, I have following questions.

1. Line 636: Fig. S1: Camera ROI is of size 1302 X 1041 pixels? Or 1041 X 1302?

2. Can you please provide statistics on the running time and memory consumption (e.g, GPU?) of the proposed framework ?

Version 1:

Reviewer comments:

Reviewer #1

(Remarks to the Author)

The authors have addressed my concerns, and I support the publication of the paper. However, I have one remaining comment, which is indicated below.

Reviewer #1 (Remarks to the Author):

1. In Fig. 2, the rightmost points ($Z \approx 450$ nm) are significantly below the CRSB for X, Y and Z. How can this be explained?

The CRLB curve is plotted at a fixed photon count (~ 1950) equal to mean photon count of the beads detected in all bead stacks. The number of beads detected in each bin and the photon counts for each bin are plotted below. In the range > 300 nm, we see a significant drop in the number of beads that were detected reliably, i.e. > 30 frames for each bead, and the average photon count of detected dots also shows a rise, especially in the last bin ~ 450 nm. I.e. they do not match the photon count used to calculate the theoretical curve.

If we include only beads from the 450 nm bin with a photon counts closer to ~ 1950 , we capture only 3 beads. For these beads the CRLB in $z \approx 56$ nm ± 19 nm, in line with the theoretical curves shown in Fig. 2. In summary, for $Z \approx 450$ nm we mainly see disproportionately bright beads which makes the CRLB expectation deviate.

I feel that the paper can be improved if this information is integrated into the manuscript or in the supplementary data. For example, the 450 nm point can be excluded due to the lack of reliable data, or the localization precision can be recalculated using only beads with photon counts around 1950. Additionally, an explanation can be included in the manuscript.

Reviewer #2

(Remarks to the Author)

The authors have addressed all my concerns.

Reviewer #3

(Remarks to the Author)

Reviewers' comments:

Reviewer #1 (Remarks to the Author):

In this paper, Karempudi et al. present a deep learning method for astigmatism-based 3D localization of chromatin loci in live *E. coli* cells. The authors characterized the PSF, fluorescence background, and system noise, and used these for training the neural network. This method provides the coordinates of fluorescently labeled molecules relative to the cell backbone, using a phase-contrast image of the cell as a reference for the coordinate system. The evaluation of the localization precision on fluorescent beads demonstrated good performance. Using this method, the authors tracked three chromosomal loci at different time scales and across the cell cycle.

The paper presents well-constructed experiments and thorough quantitative analysis. The study is overall well-executed, with the findings supporting the main conclusions. This research will likely capture the interest of a wide audience within the journal's readership. While the work is solid, I do have a few minor concerns that should be considered.

1. In Fig. 2, the rightmost points ($Z \approx 450$ nm) are significantly below the CRSB for X, Y and Z. How can this be explained?

The CRLB curve is plotted at a fixed photon count (~ 1950) equal to mean photon count of the beads detected in all bead stacks. The number of beads detected in each bin and the photon counts for each bin are plotted below. In the range > 300 nm, we see a significant drop in the number of beads that were detected reliably, i.e. > 30 frames for each bead, and the average photon count of detected dots also shows a rise, especially in the last bin ~ 450 nm. I.e. they do not match the photon count used to calculate the theoretical curve.

If we include only beads from the 450 nm bin with a photon counts closer to ~ 1950 , we capture only 3 beads. For these beads the CRLB in $z \approx 56$ nm ± 19 nm, in line with the theoretical curves shown in Fig. 2. In summary, for $Z \approx 450$ nm we mainly see disproportionately bright beads which makes the CRLB expectation deviate.

2. Lines 223-235. The description of the localization of the *ter*, *oriC*, and midway loci does not seem to accurately reflect the data in Fig. 3. The authors note that the localization of *oriC* is different from both *ter* and midway; however, in Fig. 3, the distributions of *oriC* and midway appear qualitatively similar to each other (ring-like in ZY and doublet in XY in larger cells).

We agree that the distributions for the *oriC* and midway loci are similar and our intention was to highlight the difference between them. In XY the doublet for *oriC* appears earlier than for midway and it ends up closer to the cell poles in larger cells than the midway loci, which is in line with previous observations (e.g. (Youngren et al. 2014; Cass et al. 2016; Gras et al. 2024)). Also, in YZ the midway locus is more excluded from the cell center compared to the *oriC*. We noted this distinction on lines 236-239.

3. Would it be possible to calculate the MSD for the origin and terminus and compare it with the MSD for midway?

This is clearly possible but would be a rather lengthy process, that we feel would not significantly increase the confidence in that it's possible to perform 3D localization across the cell cycle and 3D tracking, which are the main points of the paper.

4. Is there any new insight into bacterial chromatin that can be drawn from the data presented in the paper?

We believe the methods would be powerful to explore many aspects of bacterial chromatin and how proteins and RNA complexes organize the chromosome. This could, for example, involve changes in loci localization and dynamics when histone-like bacterial proteins are knocked out or when transcription is globally perturbed. Such studies have not been performed in the current work.

One conclusion about chromatin is the time scale for radial movement. The comparison between the MSD of a locus and the population distribution of the locus locations in our present work shows that the 3D diffusion of a single locus approaches the width of the population distribution after 20 min. This behavior indicates that the population distribution accurately describes the intracellular locations at the time scale of the cell cycle.

Reviewer #2 (Remarks to the Author):

The authors utilized FD-DECODE for the 3D localization and tracking of chromosomal loci in live *E. coli* cells under low SNR conditions. The manuscript is well-prepared, but many details need clarification or correction.

(1) FD-DECODE was developed to address field-dependent aberrations. In Fig. S1, it is unclear if such effects are present in such small regions (128 x 128 pixels). The rationale for training the network on these small regions instead of the full image should be explained.

We used 128x128 px regions compared to 40x40 px of the original DECODE so that we see the cell background and its variations in a few cells. Hence, the choice of region size was done to model the cell background, and not to address field-dependent aberrations. We don't model field-dependent aberrations in this work, but we use the improved implementation of the loss functions and convergence properties of FD-DECODE compared to original DECODE. This has been clarified in the main text on lines 119-125. Additionally, larger regions for training of the model require larger memory consumption.

(2) From the Methods section, it is unclear how the astigmatic PSF was generated. Was a cylindrical lens used for this purpose?

Yes, astigmatism was introduced with a cylindrical lens ($f = 500$ mm, Thorlabs LJ1144L2) in the optical path of the microscope. This was written in the Results section on line 91, but was mistakenly omitted from the Materials and Methods section. It has now been clarified on lines 385-398.

(3) Fig. S4, the astigmatic PSF is oriented at 45 degrees rather than the more commonly used 90 degrees. The reason for this orientation should be provided.

It is true that the astigmatic PSF we use is rotated 45 degrees, which was done by rotation of the cylindrical lens in the optical path. The reason for this orientation was that we aimed to minimize the bias due to alignment between the typical astigmatic PSF and the cell's long axis. The width of the bacteria is similar to the PSF and the cell background therefore changes significantly within the range of the PSF. This would cause more challenges for PSFs that are extended perpendicular to the cell long axis. The choice of PSF has now been explained on lines 97-99.

(4) Fig. S4 indicates a significant spherical aberration in the system, which results in asymmetrical z localization precision. This aberration can be corrected by adjusting the correction collar of the objective.

For all imaging experiments in the manuscript we used a Lambda-D objective that does not have a correction collar. This was the best of several objectives that we tried, including a Nikon 100X TIRF objective with a correction collar. The spherical aberration most likely comes from the tube lens. However, even with the presence of spherical aberration, our localization algorithm is still able to perform well in three dimensions. We collected stacks of fluorescent bead images to characterize the PSF of our system and trained a neural network to identify its features, including the aberrations.

(5) Fig. S5, there are well-established protocols for camera calibration. I don't know why the authors used the Accent plugin. The gain calibration result (Fig. S5A) doesn't make sense to me, which shows negative gain values. I suggest the authors use the gain value provided by the manufacturer or perform calibration following the right protocols (DOI: 10.1038/nmeth.2488; DOI:10.1038/s41467-022-30907-2).

As the reviewer suggests, Diekmann, R et al. Nature Comm. 2022 (DOI:10.1038/s41467-022-30907-2) established a good protocol for camera calibration, which is also the protocol we used in the manuscript. We cite Diekmann, R et al. on lines 563 and 718 when referring to the use of the Accent plugin (automated camera characterization via electron noise tool) which was described in the paper. The negative values are reported by the Accent plugin.

Reviewer #3 (Remarks to the Author):

Localizing chromosomal loci in the Escherichia coli cell cycle in 3D is challenging due to the low signal-to-background ratio, and the structured background is essential for the localization and the generation of training data. To overcome this challenging, this manuscript proposed and developed a field-dependent neural network based on Fu et al. (2022)'s field-dependent DeepLoc algorithm. The contributions of this manuscript include simulating ground truth training data and 3D localization with high localization accuracy. However, I have following questions.

1. Line 636: Fig. S1: Camera ROI is of size 1302 X 1041 pixels? Or 1041 X 1302?

The camera ROI is 1041 X 1302 pixels. The size on line 636 was a typo which has now been corrected.

2. Can you please provide statistics on the running time and memory consumption (e.g, GPU?) of the proposed framework ?

Runtime for localizing molecules on an image stack of size 300x1041x1302 is ~90 s and GPU memory consumption is 5 GB on NVIDIA RTX 4090 and CPU (i5-9600K @3.70GHz 6 cores) memory consumption is ~5 GB.